# A Cross-Reconstruction Method for Step-Changed Runoff Series to Implement Frequency Analysis under Changing Environment

**DOI:** 10.3390/ijerph16224345

**Published:** 2019-11-07

**Authors:** Jiantao Yang, Hongbo Zhang, Chongfeng Ren, Zhengnian Nan, Xiaowei Wei, Ci Li

**Affiliations:** 1School of Environmental Science and Engineering, Chang’an University, Xi’an 710054, China; 2017129011@chd.edu.cn (J.Y.); rchf@chd.edu.cn (C.R.); 2016229028@chd.edu.cn (Z.N.); 2017129014@chd.edu.cn (X.W.); 2017229012@chd.edu.cn (C.L.); 2Key Laboratory of Subsurface Hydrology and Ecological Effect in Arid Region, Ministry of Education, Chang’an University, Xi’an 710054, China

**Keywords:** hydrology, data expansion, stationarity, EMD, frequency analysis, environmental change

## Abstract

The stationarity of observed hydrological series has been broken or destroyed in many areas worldwide due to changing environments, causing hydrologic designs under stationarity assumption to be questioned and placing designed projects under threat. This paper proposed a data expansion approach—namely, the cross-reconstruction (CR) method—for frequency analysis for a step-changed runoff series combined with the empirical mode decomposition (EMD) method. The purpose is to expand the small data on each step to meet the requirements of data capacity for frequency analysis and to provide more reliable statistics within a stepped runoff series. Taking runoff records at three gauges in western China as examples, the results showed that the cross-reconstruction method has the advantage of data expansion of the small sample runoff data, and the expanded runoff data at steps can meet the data capacity requirements for frequency analysis. In addition, the comparison of the expanded and measured data at steps indicated that the expanded data can demonstrate the statistics closer to the potential data population, rather than just reflecting the measured data. Therefore, it is considered that the CR method ought to be available in frequency analysis for step-changed records, can be used as a tool to construct the hydrological probability distribution under different levels of changing environments (at different steps) through data expansion, and can further assist policy-making in water resources management in the future.

## 1. Introduction

Hydrological statistics are widely employed in water resources planning and management around the world currently [1], in which the stationarity assumption of a hydrological time series is fundamental, influencing the effectiveness of water resources management [2]. Stationarity is predicated upon an assumption that the generating process is in equilibrium around an underlying mean and that variance remains constant over time [3]. With the climate change and human activities in river basins, water management professionals have been counseled to abandon historical assumptions of stationarity in the natural systems governing surface water replenishment [3]. For example, Milly et al. [4] claimed that stationarity is dead and could not be revived because of the substantial anthropogenic change of the Earth’s climate altering the means and extremes of precipitation, evapotranspiration, and rates of discharge of rivers. After that, the claim was questioned by some scholars, and some issues that can be discussed were put forward regarding whether stationarity is indeed dead or now has only a secondary statue. Matalas [5] stated that stationarity is always a part of the composition of a time series, explicit or otherwise. Stationarity may be overlooked, but it remains alive and well. Furthermore, the operational assumption of hydrologic stationarity used extensively in water management should not be discarded without further study [5]. Chung et al. [2] reported that the fundamental cause of this climatic change is the warming temperature, and further noted no apparent change in the runoff for the last century. Although it is uncertain as to whether stationarity is dead or alive, it is well known that human activities and climate change have a significant impact on runoff and other hydrological processes [6,7], which implied how to use the nonstationary runoff records to implement hydrological frequency analysis.

It is often seen that the measured runoff records may be different from the runoff under certain environments, reflecting the significant nonstationarity, due to some changing influences such as the direct human water withdrawal from the river channel [8]. Under the traditional hydrological frequency analysis, the reconstruction of a stationary runoff series (natural runoff) should be done for frequency analysis by combining the measured series with the investigation results of water withdrawal for various industries. However, it should be noted that the reconstructed (natural) runoff series are often not seen in practice at the gauges, only reflecting a theoretical value or the hydrological process in areas without human interference, and being unable to provide guidance for engineering design. Unlike this, the measured runoff records essentially present an inflow status at the observed gauge or available surface water resources, which is important evidence for engineering design under current conditions. Unfortunately, these records are often nonstationary (such as trend or step changes), and difficult to use for frequency analysis. Thus, it is very vital to construct the probability distribution and carry out hydrological frequency analysis under the conditions of the current stationary environment, which has a greater significance to regional water resources management than ‘natural’ runoff.

To date, a series of studies on the probability distribution of nonstationary data have been put forward, in which the most popular approach is the frequency analysis with covariates method [9]. The covariates approach incorporates the covariates into the distribution parameters [10,11]; generally, two distributions are employed. Some studies have provided frequency models for taking nonstationarity into account, such as the generalized extreme value model (GEV) for block maxima [12,13] and the peak-over-threshold (POT) modeling approach [14,15]. Otherwise, Cannon et al. [16] have used linear quantile regression to estimate nonstationary extreme events. Sarhadi et al. [17] established a methodological approach for mapping the probability of flooding, linking Geographic Information System (GIS) techniques with frequency analysis. Brodie [18] has used the rational equation and Monte Carlo method to provide an independent frequency determination of the peak discharge. Nasri et al. [9] proposed using B-spline quantile regression for hydrologic frequency analysis. In addition, some researchers adopted the copula function proposed by Sklar [19] for multivariate frequency analysis in hydrology [20,21], especially drought frequency analysis. Here, it is clarified that the above approaches for frequency analysis also need a certain amount of data. They could be helpless when data are lacking.

The aim of the study was to develop a new method—namely, the cross-reconstruction (CR) method—that can provide a frequency analysis of a step-changed measured runoff series. It is based on the empirical mode decomposition (EMD) method, aiming to expand the small sample data on each step to make it meet the requirement of data capacity for frequency analysis. In addition, the step-change presents the stationarity of measured data at each step; thus, its statistical property can be considered to reflect the condition of runoff production under the current environment. It is expected that the distribution of runoff series at different steps will be established, and the design value at typical frequencies will be able to be estimated corresponding to various environment conditions (including human water withdrawal), if the cross-reconstruction method could solve the data expansion of a small measured sample for meeting the requirement of data capacity in frequency analysis effectively.

## 2. Materials and Methods

### 2.1. Step-Changed Runoff Series and Environment Scenarios

It is well known that the changes in many things or systems in the world are stepped or phased; that is, after a period of time, they jump to another state and exist steadily. From the point of view of systematics, this phenomenon can be interpreted as the open equilibrium state of the system. When strongly influenced by some factors, the system is nonstationary, and the external characteristics (such as observed data series) show nonstationary. After a time, the system adjusts and re-establishes a new equilibrium state, resulting in a second open equilibrium state. Reflecting the hydrological system, it can be believed that the step-changed hydrological series is the representation of the change of the hydrological system and that different steps are the stationarity state of the hydrological system at different stages. In a stationarity state, it can be assumed that the external environment of the hydrological system is stationarity; that is, the impact of climate change and human activities remains relatively stationary. If the measured hydrological series are nonstationary, it can be considered that the system states of multi-representation are different. At this time, it is not necessarily appropriate to unify the frequency analysis. Therefore, this paper will study the statistical characteristics of different states of the hydrological system by the sample expansion method and analyze the evolution trend and characteristics of the statistical characteristics of hydrological data in different states. The specific process is shown in Figure 1.

### 2.2. Pettitt Test and Iterative Cumulative Sum of Squares for Segmentation

Segmentation is fundamental to data cross and reconstruction. In this process, Pettitt test and iterative cumulative sum of squares were employed to detect change points in the mean and variance of a runoff series, by which the measured runoff series were segmented.

#### 2.2.1. Pettitt Test

Pettitt’s test is a widely used nonparametric test developed by Pettitt [22] to detect change points in a time series with continuous data. It can detect the location of change points in the mean when the exact time of the change is unknown, and can give the associated significance probability. Pettitt’s test considers a series of random variables *X_t_* (*t* = 1, …, N). It tests the null hypothesis, *H*_0_: The *X_t_* variables follow one or more distributions that have the same location parameter (no change), against the alternative hypothesis: a change point exists. The test uses the *U_t,N_* value, which tests whether two sample sets, *x*_1_, ..., *x_t_* and *x_t_*_+1_, ..., *x_N_*, are from the same population. The test statistic *U_t,N_* is given by Gao et al. [23]:(1)$$U_{t,N}=U_{t-1,N}+\sum_{i=1}^{N}sgn(x_{t}-x_{i}),\;t=2,\;L,\;N$$
where *sgn* is a sign function and *sgn*(*x_t_* − *x_i_*) = 1 if *x_t_* − *x_i_* > 0, 0 if *x_t_* − *x_i_* = 0, and −1 if *x_t_* − *x_i_* < 0. The test statistic *U_t,N_* counts the number of times that a member of the first sample exceeds a member of the second sample. The maximum of the absolute values, |*U_t,N_*|, gives the position of a possible change point if 1 ≤ *t* < T. The statistic and the associated significance probability (*p*) used in the test are given as follows:(2)$$k_{t,N}=\max_{1\leq t\leq N}\left|U_{t,N}\right|$$
(3)$$p≅1-2\exp(\frac{-6k_{t,N}^{2}}{N^{3}+N^{2}})$$
the change point of the series is located at *k_t,N_* if the significance probability *p* is equal to or greater than 0.95.

#### 2.2.2. Iterative Cumulative Sum of Squares

The iterative cumulative sum of squares (ICSS) algorithm has been proposed by Inclan and Tiao [24] to detect multiple breakpoints in a variance of a time series. The algorithm, including a centered cumulative sum of squares and the iterative procedure, is widely used to detect multiple shifts in climate and hydrological data. Therein, the centered cumulative sum of squares is regarded as the test statistics *D_k_* to estimate the number of changes and the point in time of variance shifts. The specific steps are as follows:(4)$$D_{k}=\frac{c_{k}}{c_{N}}-\frac{k}{N},\;D_{0}=D_{N}=0$$
(5)$$C_{k}=\sum_{i=1}^{k}{\left[x(i)\right]}^{2}\;1<k<N$$
where *N* is the length of the series *x*(*i*), *C_k_* is the cumulative sum of squares of independent random variables in series *x*(*i*) with a mean of 0 and variances of $\sigma_{i}^{2}$, and *C_N_* is the sum of squared errors in the whole time series. According to the *D_k_* value, the sudden changes in variance can be identified in the time series as follows:(a)If *D_k_* oscillates around zero, it represents that there is no change in the variance over the whole period;(b)If *D_k_* departs from zero, it means that there are one or more shifts in variance;(c)If the maximum of $\sqrt{N/2}\left|D_{k}\right|$ exceeds the boundary values that are obtained from the asymptotic distribution of *D_k_*, assuming constant variance, a significant shift in variance occurs at *k*. The 5% significant level is selected in the study, and ±1.358 is the asymptotic critical value.

### 2.3. Empirical Mode Decomposition

Empirical mode decomposition (EMD) was proposed by Huang et al. [25], and is considered a signal analysis method to analyze the nonlinear and nonstationary data [26]. Its main idea is to sift the nonlinear and nonstationary time series data by Hilbert–Huang transform (HHT) until the series can be reconstructed by several decomposed stationary data series as intrinsic mode functions (IMFs) and residual components [27]. This sifting process is described as follows:
(a)Identify all local extrema containing the maxima and minima of the original time series X(t), and then fit all the local extrema with a cubic spline function to produce the lower and upper envelopes, $e_{min}\left(t\right)$ and $e_{max}\left(t\right)$, respectively.(b)Compute the mean of the envelopes by $m\left(t\right)=\left[e_{max}\left(t\right)+e_{min}\left(t\right)\right]/2$.(c)Obtain the new series by h(t)=X(t)−m(t).(d)Check whether h(t) is a nonstationary series; if yes, the above procedures must be repeated k times until m(t)=0. If not, h(t) is regarded as the first intrinsic mode function (IMF) as $I_{1}\left(t\right)$. The first IMF represents the highest frequency component of the original series.(e)Calculate the margin series $r_{1}\left(t\right)=X\left(t\right)-I_{1}\left(t\right)$, and the second IMF can be obtained from it. Such a procedure needs to be repeated until the last margin series $r_{n}\left(t\right)$ is a monotone series and can’t be decomposed further. At this point, the decomposing is finished, and *n*-1 IMFs and a residual component are obtained where the residual $r_{n}\left(t\right)$ represents the mean trend of the original series. If calculating the summation of all IMFs and the residual, the original series X(t) can be reconstructed as follows:
(6)$$X\left(t\right)=\sum_{i=1}^{n}I_{i}\left(t\right)+r\left(t\right)$$

### 2.4. Kolmogorov–Smirnov (K-S) Test

The two-sample Kolmogorov–Smirnov (K-S) test is a non-parametric hypothesis test that quantifies a distance between the empirical distribution function of the sample and the cumulative distribution function of the reference distribution, or between the empirical distribution functions of two samples [28]. The test uses as input two sample data vectors generated from probability distributions, and the test statistic *D* is formulated as:(7)$$D=\underset{x}{\sup}\left|F_{n_{1}}\left(x\right)-G_{n_{2}}\left(x\right)\right|$$
where $F_{n_{1}}\left(x\right)$ and $G_{n_{2}}\left(x\right)$ are the empirical distribution function of the first and the second sample respectively, and *sup* is the supremum function. The null hypothesis is rejected at the significance level *α* of the hypothesis test if:(8)$$D>c\left(\alpha \right)\sqrt{\frac{n+m}{nm}}$$
where *n* and *m* are the sizes of the first and second sample, respectively, and the value of *c*(*α*) for each significance level of *α* can be retrieved from Miller [29]. The threshold value of differentiation is 0.05 in this paper according to the significance level of the K-S test.

### 2.5. Cross and Reconstruction

#### 2.5.1. Cross Procedure

It is generally mentioned in relevant hydrological researches that the IMFs decomposed by EMD mainly include the periodic change caused by climate and the random change caused by climate and human impacts. Residual contains trend-influenced information on climate change and human activities. When the residual is stable, it can be considered that the hydrological system is in a relatively stable situation. At this time, the decomposed IMFs are tested by the K-S test. If the test is met, indicating that the segmented series of verified IMFs are with similar central frequency, they could be considered to be driven by similar climatic factors. Thus, the verified IMFs can be implemented by the data component cross procedure.

The cross procedure comes from the genetic crossover in genetic algorithms (GAs) simulating Darwin’s genetic selection and the natural selection process of biological evolution. In this paper, the cross was carried out among the segmentations in the verified IMFs, shown in Figure 2, and *n^n^* (*n* is the number of segmentation) data samples can be obtained.

#### 2.5.2. Runoff Reconstruction

The reconstructed new series can be obtained by combining the IMFs with similar central frequency and adding a steady residual. Figure 3 shows the principle of data reconstruction, and the sifting procedure can be described as follows:
(a)The original runoff series are segmented based on change points in mean and variance.(b)Each segmented series should take the logarithm of an extended data series with the neural network for mitigating the boundary error [30], which is then decomposed by EMD. At this moment, multiple intrinsic mode functions (IMFs) and a residual trend term (Residual) of each segment can be obtained.(c)All the IMFs are done with the similarity detection by the K-S test. The similar IMFs are combined and superimposed. Figure 3 shows that each segment can be reconstructed by $IMF1+IMF2+IMF4+Res+C_{4}^{1}$ when the decomposed IMFs of each segment are similar, and the original series can eventually be expanded to a new sample with the size of $C_{4}^{1}\times C_{4}^{1}\times C_{4}^{1}\times C_{4}^{1}$.

## 3. Method Validation

To verify the applicability of the component reconstruction method, a set of mathematical tests were designed to detect the performance of the method. Assuming that a group of random series obeys normal distribution, the series was segmented uniformly, and the segmented series was amplified by the component reconstruction method. By comparing the mean value, the variance of the synthetic series, and the random series, the applicability of the CR method was verified.

Firstly, a group of random series subject to normal distribution X~N (0,1)—namely, the mean value is 0, the variance is 1, and the series length is 5000—were randomly generated as the population sample. In addition, using the resampling method to extract 400 samples from the population, every sample was evenly divided into four segments with a length of 100. Furthermore, the segments were decomposed by EMD, and the newly synthesized series was obtained by the cross-reconstruction method. The above process was recorded as an experimental process. For a more reliable result, 50 experiments were conducted, and the results were analyzed. The mean and variance statistics of the reconstructed synthetic series and extracted samples in 50 experiments and the statistics of the population are shown in Figure 4. According to the graphs, it can be known that the mean and variance of the 50 experimental synthetic series were close to that of the population; however, the mean and variance of the 50 extracted samples were quite different from it, implying that the reconstructed synthetic series were reasonable and can be used to expand the data sample.

## 4. Application

### 4.1. Study Area and Data

#### 4.1.1. Study Area

The Wei River, the largest tributary of the Yellow River, originates from the north of the Wushu Mountain in the southwest of Weiyuan County, Gansu province, China. It flows through Gansu, Ningxia, and Shaanxi provinces and runs into the Yellow River at Tongguan, Shaanxi province [31]. The river has a total length of 818 km, and a basin area of 134,800 km^2^, as shown in Figure 5 [32,33]. In general, the basin topography fluctuates greatly, and the altitude decreases from the highest northwest mountainous area to the lowest Guanzhong Plain in the southeast and southern area, with the largest elevation difference over 3000 m [34,35]. The whole basin belongs to the continental monsoon climate, with the annual mean temperature of 7.8–13.5 °C, precipitation of 450–700 mm, and pan evaporation of 1000–2000 mm [36]. The average annual natural runoff of the river is 10.4 billion m^3^, contributing 17.3% of the total discharge of the Yellow River. The drainage system of the Wei River basin is well developed; there are numerous water conservancy projects and irrigation projects on its mainstream and tributaries, triggering the nonstationary streamflow. Therefore, reconstructing the stationary runoff series in the Wei River basin is very significant to frequency analysis and can provide a scientific basis for water resources assessment, water conservancy project construction, and basin economic development.

The Tuwei River basin is located in the northern part of the Loess Plateau (approximately 109°45′–110°35′ N, 38°10′–39°10′ E) [37], and is also the first-order tributary of the Yellow River. It originates from Gongbohaizi in Yao Town, Shenmu County, Shaanxi province. It flows through Shenmu, Yuyang, and Jia counties in Yulin city. The river has a total length of 140 km and a basin area of 3294 km^2^ (Figure 5). The average annual precipitation is 402 mm (1971–2011), the average annual runoff is 0.435 billion m^3^, and the pan evaporation is 1853 mm (1971–2004) [38]. In the basin, the Shuidong canal, the Honghua canal, the Yongxing canal, and other irrigation channels have been built in past years. Due to the influence of these human-induced activities, the observed runoff data in Tuwei River basin has not been able to be directly used for water conservancy project construction and water resources assessment. Therefore, it is necessary for the Tu Wei River basin to deal with the nonstationary measured runoff.

#### 4.1.2. Data

The data used in this paper include the monthly measured runoff at Xianyang hydrological gauge (Figure 5) on the mainstream of the Wei River and Gaojiabao hydrological gauge (Figure 5) on the Tuwei River. The runoff data series of Xianyang gauge is from 1956 to 2000, and that of the Gaojiabao gauge covers 1976–2010. Through the data at two gauges, the hydrological characteristics of the two different regions or basins can be represented, which also have the advantage of verifying the applicability of the method in various regions. The data was obtained from hydrological manuals published by the Hydrological Bureau of the Yellow River Conservancy Commission.

### 4.2. Results

#### 4.2.1. Segmentation

The Pettitt method was used to detect the mean change points as a group boundary in this study. Through the approach, three change points were identified in a measured runoff series at Xianyang gauge, occurring in 1968, 1985, and 1993, respectively. Similarly, three change points were detected at Gaojiabao gauge, which were located in 1984, 1993, and 2003, respectively. In addition, the ICSS method was applied to find change points in variance. The results showed that a change point in variation occurred in 1985 at Xianyang gauge, and the change in variance at Gaojiabao gauge occurred in 2003. The reason for using the Pettitt and ICSS methods to detect the change points is that these detected change-points in mean and variance perhaps changed the original probability density distribution of the series, which should be considered to segment. When the cross-reconstruction (CR) method was used, it was necessary to select the data series with the same length from each segmentation, aiming to ensure that the segmented data series can be effectively crossed and reconstructed when using the CR method. According to the locations of change points in mean and variance, the runoff series at Xianyang gauge was divided into four segments: 1956–1962, 1969–1975, 1987–1993, and 1994–2000, and the segmentation results are shown in Figure 6. The series at Gaijiabao gauge was divided into four segments: 1976–1982, 1985–1991, 1994–2000, and 2004–2010, and the results are demonstrated in Figure 7. It is noted that the length of each data segment at Xianyang and Gaojiabao gauges was seven years, including 84 months of runoff data, approximately. Since the annual runoff series is too short in segments, the monthly runoff data of each segment was selected for decomposition in this study.

It can be found from Figure 6 and Figure 7 that the statistical characteristics of various data segments were different at Xianyang and Gaojiabao gauges, implying the segmentation of the runoff series was reasonable by the Pettitt and ICSS approaches and can be used for the next runoff data expansion.

#### 4.2.2. Decomposition

The EMD method was employed to decompose the measured monthly runoff series at Xianyang and Gaojiabao gauges. The results at Xianyang gauge are shown in Figure 8. The results at Gaojiabao gauge are demonstrated in Figure 9. From the figures, it can be known that the decomposed serial components become stable and periodic with the increase of decomposition times. Each segmented measured runoff series can be decomposed into five intrinsic mode functions (IMFs) and one residual. Using Mann–Kendall (M-K) on the residuals, the trend test results at Xianyang gauge showed that the statistics |Z| of the residual in each segment were 0.93, 0.72, 1.37, and 0.84 respectively, which were all less than 1.96, indicating that the trend of the residuals in segments are not significant. In the same way, the trend test results at Gaojiabao gauge indicated that the statistics value of each segment were 1.26, 1.47, 0.81, 0.63, and all were less than 1.96, implying the trend changes in segments are also non-significant. Thus, it can be considered that the residuals are relatively stable in segments at two gauges, and can be used for the reconstruction in the C-R method.

#### 4.2.3. Kolmogorov–Smirnov (K-S) Test

The Kolmogorov–Smirnov (K-S) test was used between IMFs. The K-S results are shown in Figure 10.

Figure 11 shows the similar IMFs at Xianyang gauge. It can be seen from the graph that there are similarities between the IMF1s in 1969–1975 and 1987–1993, IMF2s in 1969–1975 and 1987–1993, IMF3s in 1956–1962 and 1987–1993, IMF4s in 1956–1962 and 1969–1975, and IMF5s in 1987–1993 and 1994–2000. The results at Gaojiabao gauge are displayed in Figure 12. It was found from the graph that there are similarities between IMF1s in 1985–1991 and 1994–2000, IMF2s in 1976–1982 and 1985–1991, IMF3s in 1985–1991 and 2004–2010, IMF4s in 1976–1982 and 1994–2000, and IMF5s in 1985–1991 and 1994–2000. According to the results, the cross will be implemented between similar IMFs for further reconstruction.

#### 4.2.4. Cross and Reconstruction

It can be known from the K-S test results that five pairs of similar IMFs at Xianyang gauge would be crossed and reconstructed with the residual at different segments respectively so that a 2^5^ synthetic runoff data series can be obtained, as shown in Figure 13. Each synthetic series contained 2688 data points in total, and they were compared with the measured series by the scatter diagram. In the figures, Figure 13a–d is the combined scatter diagram of the synthetic series point and the measured series point, and Figure 13f–i is the amplified diagram at the bottom area of Figure 13a–d. It can be found from the figures that the synthetic series points completely covered the measured points, implying the performance of the CR method. In addition, the box figures of the synthetic series and the measured series for parameter comparison are shown in Figure 13k–n. Figure 13e,o present the whole runoff series from 1956 to 2000 and its statistics, and Figure 13j is the amplified diagram at the bottom area of Figure 13e. From the figures, it can be found that the synthetic series covered all the original measured series; however, the statistical parameters of the synthetic series in segments were quite different from those of the original series due to the nonstationarity of the original data series. At Gaojiabao gauge, five similar IMFs would be crossed and reconstructed with the residual at different segments, respectively. The synthetic runoff data series are shown in Figure 14a–d. It can be seen from the graph that the synthetic series also covered the measured series. The parameter comparison was demonstrated by the box figures in Figure 14f–i. Figure 14e,j presents the whole runoff series from 1976 to 2010 and its statistics. From the figures, it can be also found that the synthetic series covered all the original measured series at Gaojiabao gauge, although some original data was not used for data expansion. The statistical parameters of the synthetic series in segments were also certainly different from those of the original series.

As shown in Figure 13k–n and Figure 14f–i, the minimum value of the synthesis series is very close to that of the measured series, and the maximum value of the synthesis series is a little larger than that of the measured series. In addition, the values of 25%, 50%, and 75% of the synthetic series are close to those of the measured series, except for the series in 1994–2000 at Gaojiabao gauge, indicating that the synthetic series is similar to the measured series.

### 4.3. Discussion

The purpose of data expansion is to improve the frequency analysis of small-sample hydrological data. Thus, the Pearson type III (P-III) distribution was employed to detect the difference between the synthetic and measured series. The statistical parameters of P-III distribution at Xianyang gauge are shown in Figure 15. From the figure, it can be seen that the P-III parameters of each segment in the synthetic series are very close to those of the measured runoff series, implying that the CR synthetic series is consistent with the measured series. Drawing P-III curves of the synthetic and measured series in Figure 15, it was found through the comparison between the two curves that the distribution of each segmented synthetic and measured series has good similarity, but the fitting curves have a certain deviation on the return period of extreme wet events (once-in-a-century and millennium). Due to the length of the measured series data being relatively short, there is a lack of extreme wet events, and thus the frequency values would usually be uncertain through the fitting curve. However, the synthetic series contains more data, closer to the data population, and so the frequency values obtained by the P-III curve fitted by the synthetic series will be more reliable than those from the measured data and are suitable for use in engineering design.

The P-III frequency curves and statistical parameters of segmented synthetic and measured series at Gaojiabao gauge are shown in Figure 16. Similarly, the results show that the parameters of the segmented synthetic series were very close to those of the measured series. The P-III curves of each segment in the synthetic and measured series have a good similarity. The synthetic data points and the measured data points were well distributed around two P-III curves, but similar to those at Xianyang gauge, there were deviations in the extreme values (wet and dry).

Focusing on Figure 15 and Figure 16, it can be found that the synthetic series to different steps were obtained by the cross-reconstruction method, representing generated runoff under various environment scenarios. When the runoff series have step changes, the CR method will expand the small sample hydrological data; furthermore, it also fits a probability density function for hydrological frequency analysis, which is the advantage of this approach in solving how to build the distribution function based on a few data. Meanwhile, the comparison results with the measured data indicated that the distribution parameters of the synthetic series were very close to those of the measured runoff series, and were able to cover the distribution of the measured data completely. In addition, the synthetic series contained more information on potential data populations embedded in historical measured data and further provided more possibilities, especially in the extreme wet value, making it easier to draw the probability density curves. Thus, the CR method is suggested when the hydrological data capacity does not meet the requirement to implement frequency analysis. In particular, when the extreme runoff value is needed, but can not be obtained from the limited measured data, the CR method is able to offer a significant guidance in practical work. However, it should be noted that the frequency analysis results by the CR method only serve as a reference for some important projects, and more observations are necessary.

In this study, two hydrological gauges in different basins were selected for the study. However, owing to the two basins being relatively close, the result could be not convincing. Thus, Xiangtan gauge (112°93′ E, 27°83′ N) at Dongting Lake in southeast China was selected as the third case study for further verification, which has an annual runoff series covering 1959–2014 that was selected for the calculations. The change-point test result showed that the change points in the series appeared in 1988. Then, the annual runoff series were divided into two segments as 1962–1987 and 1989–2014. Next, the CR method was used to expand the data in two segments, and the synthetic runoff data series are shown in Figure 17 with measured data. It can be seen from Figure 17a,b that the synthetic series covered the measured series as well. The parameter comparison was demonstrated in Figure 17d,e where the minimum value of the synthesis series is very close to that of the measured series, and the maximum value of the synthesis series is a little larger than that of the measured series. In addition, the values of 25%, 50%, and 75% of the synthetic series are close to these of the measured series, indicating that the synthetic series is similar to the measured series. Figure 17c,f present the whole runoff series from 1945 to 2013 and its statistics. From the figures, it can be found that the synthetic series covered all the original measured series, and more high flow and low flow occurred in the synthetic series than in the original series.

The P-III frequency curves and statistical parameters of the segmented synthetic and measured series at Xiangtan gauge are shown in Figure 18. Similarly, the results showed that the parameters of the segmented synthetic series were very close to those of the measured series, and the P-III curves of the segmented synthetic and measured series have a good similarity. In addition, it is seen that the segmented synthetic series provides more data points to help determine the P-III curves.

## 5. Conclusions

In this paper, a new data expansion approach, namely the cross-reconstruction (CR) method, is proposed for frequency analysis of step-changed measured runoff series. Its advantage is that it can build the probability density function on different steps through data expansion, solving the hydrologic design of small sample hydrological data. In addition, the results of frequency analysis on a different step can present hydrological features under different environmental scenarios and are expected to serve water resources assessment and management under a changing environment. Through the analysis and discussion, the conclusions are as follows:
(a)The cross-reconstruction method can effectively implement the data expansion and obtain the synthetic series meeting the requirement of data capacity for frequency analysis, which is closer to the population.(b)The synthetic series obtained can cover all the measured data and contain more information about the potential data population embedded in historical measured data, reflecting more possibilities in the future. The method has achieved satisfactory performance in a different watershed, implying that it has good applicability.(c)The synthetic series presented similarity with the measured data in distribution parameters and probability curves, and has an advantage in its extreme wet value, contributing to drawing reasonable probability density curves and obtaining an accurate design value. Thus, the CR method is suggested to offer a significant guide in practical work when the extreme runoff value is needed, but it can not be obtained from the limited measured data.(d)The synthetic series is generated at different steps, and its distribution characteristics can present the changes of runoff under different environment scenarios, including climate and social development situations. These advantages are significant for calculating the strategy of water resources management under a changing environment.

Although the CR method has good performance on data expansion and can be used for hydrological frequency analysis, it should be emphasized that the frequency analysis results by the CR method only serve as a reference for some important projects, and more observations are still necessary.

## Figures and Tables

**Figure 1 ijerph-16-04345-f001:**
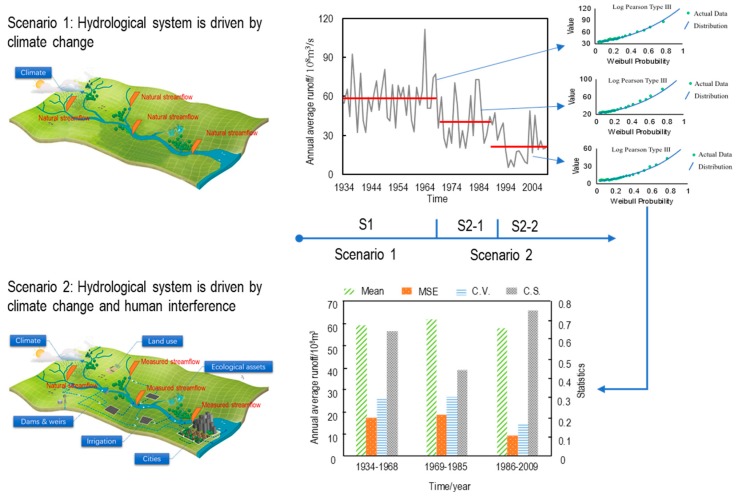
Step-changed runoff series and environment scenarios. In Figure 1, the Mean represents the mean value of each segment, MSE is the mean square error, C.V. is the coefficient of variation, and C.S. presents the deviation coefficient of each segment.

**Figure 2 ijerph-16-04345-f002:**
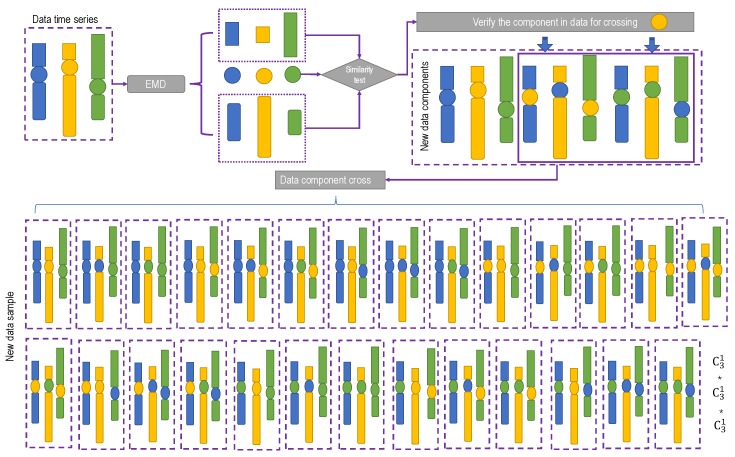
Data component cross procedure. In the figure, the rectangle, rounded rectangle, and circle represent various intrinsic mode functions (IMFs) respectively (IMF1, IMF2, IMF3, ….) decomposed by empirical mode decomposition (EMD) in each segment. The colors indicate the different segments. The cross is implemented between the same IMF components of two segments meeting the similarity test.

**Figure 3 ijerph-16-04345-f003:**
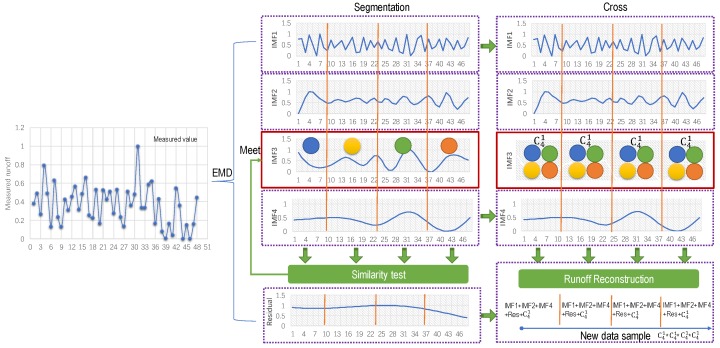
The process of component reconstruction.

**Figure 4 ijerph-16-04345-f004:**
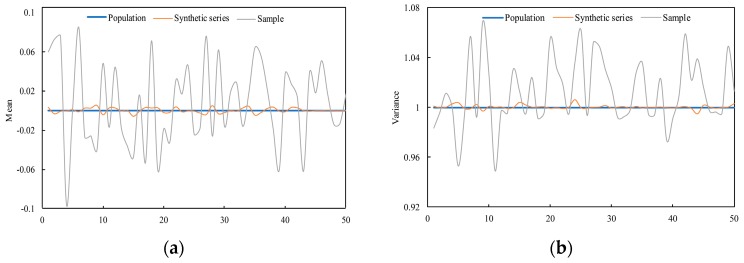
Changes in the mean (**a**) and variance (**b**) of 50 experimental samples, a synthetic series, and the population.

**Figure 5 ijerph-16-04345-f005:**
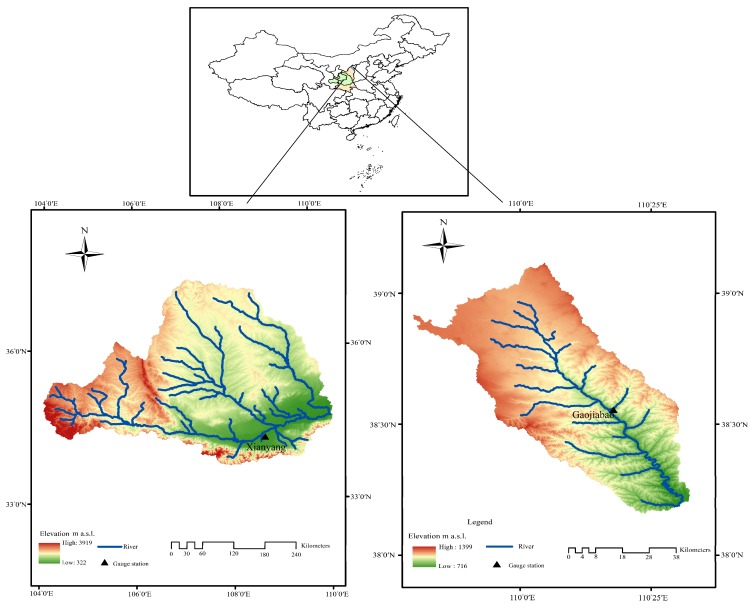
Study area and hydrological gauges.

**Figure 6 ijerph-16-04345-f006:**
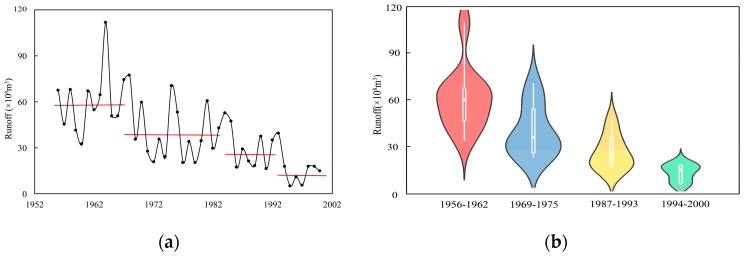
Segmentation of runoff series and their violin figures at Xianyang gauge. (**a**) the runoff series of each segment; (**b**) the violin figures of each segment.

**Figure 7 ijerph-16-04345-f007:**
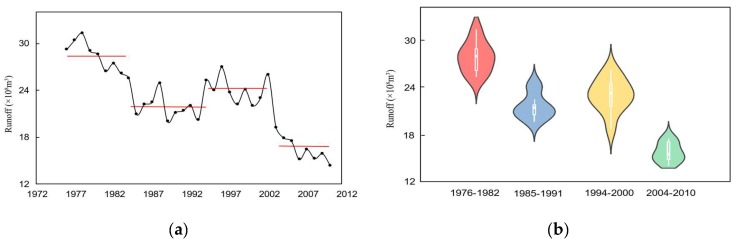
Segmentation of runoff series and their violin figures at Gaojiabao gauge. (**a**) the runoff series of each segment; (**b**) the violin figures of each segment.

**Figure 8 ijerph-16-04345-f008:**
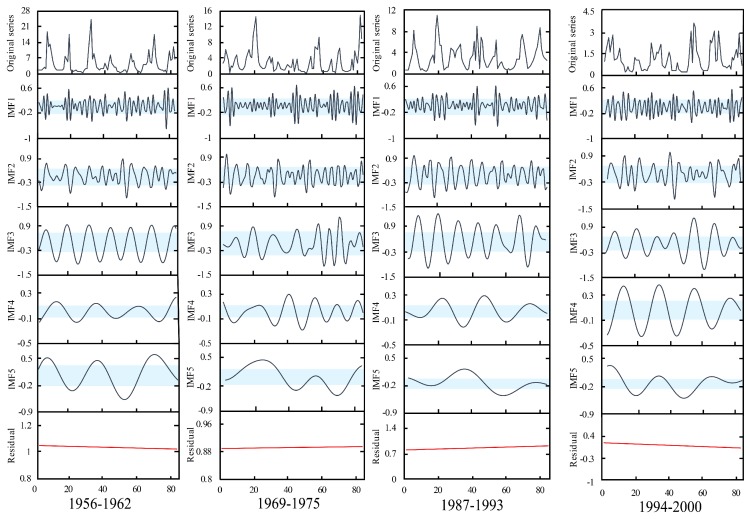
Segmented decomposed series of measured monthly runoff at Xianyang gauge.

**Figure 9 ijerph-16-04345-f009:**
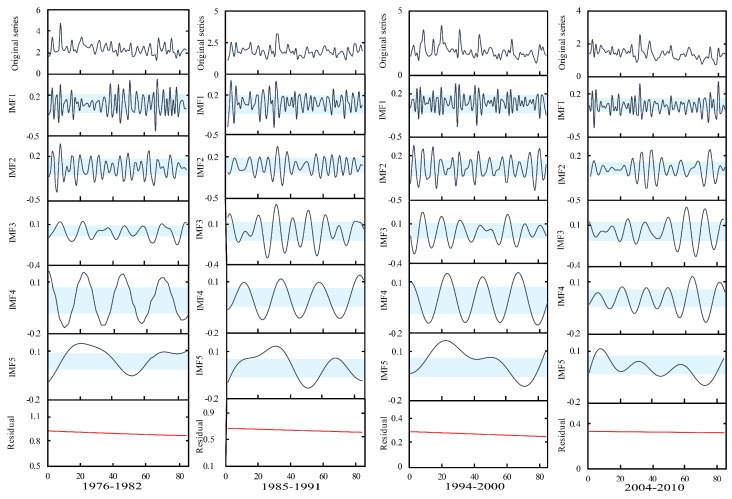
Segmented decomposed series of measured monthly runoff at Gaojiabao gauge.

**Figure 10 ijerph-16-04345-f010:**
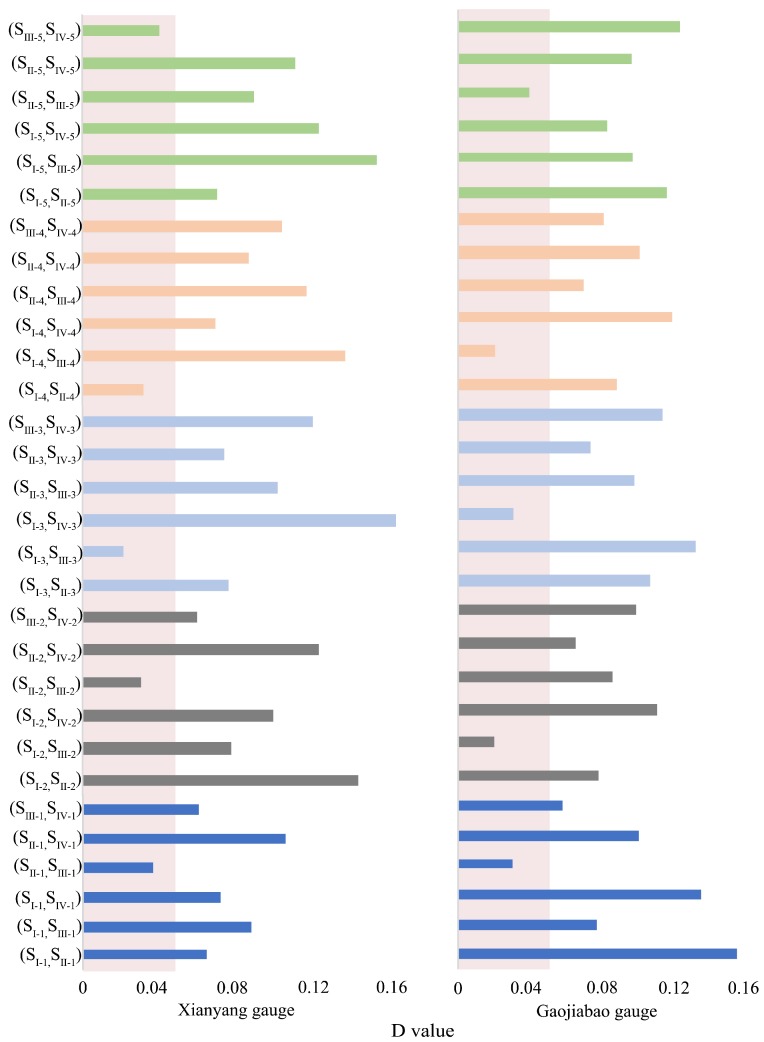
Kolmogorov–Smirnov (K-S) test results. S_I-1_ denotes the IMF1 decomposed from the first segment, S_II-1_ denotes the IMF1 decomposed from the second segment, and the red areas demonstrate confidence intervals of 0.05, by which the significant similar IMFs can be identified.

**Figure 11 ijerph-16-04345-f011:**
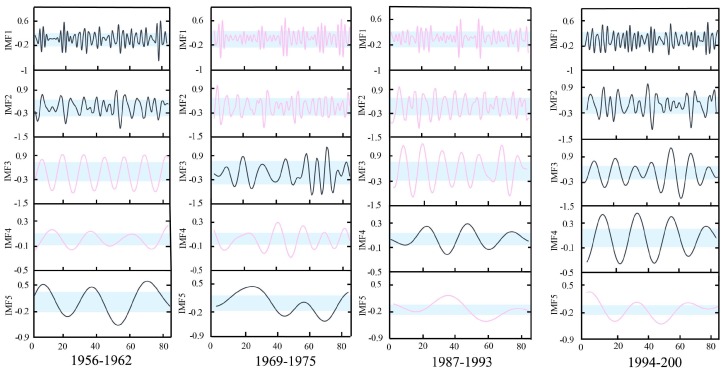
Similarity of segmented IMFs at Xianyang gauge. The pink lines in segments represent similar IMFs detected by the K-S test, and the blue areas demonstrate change intervals of 25%–75%.

**Figure 12 ijerph-16-04345-f012:**
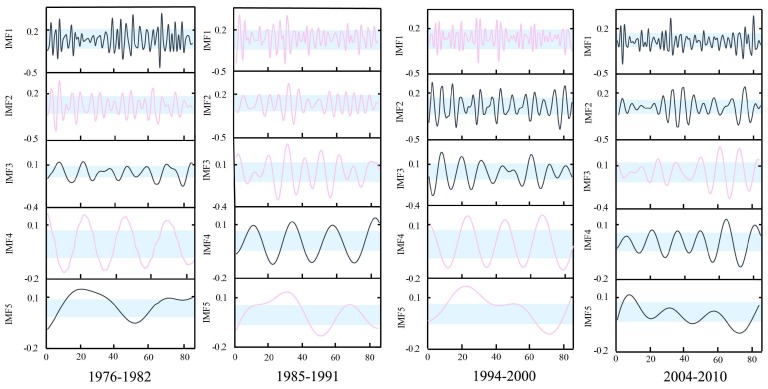
Similarity of segmented IMFs at Gaojiabao gauge. The pink lines in segments represent similar IMFs detected by the K-S test, and the blue areas demonstrate change intervals of 25%–75%.

**Figure 13 ijerph-16-04345-f013:**
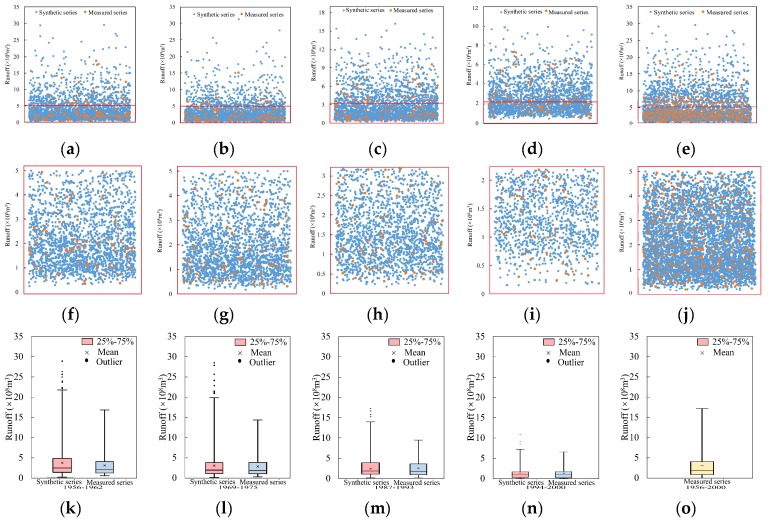
Scatter diagram (**a**–**d**) and boxplots (**k**–**n**) of the synthetic series and measured series in four segments at Xianyang gauge; (**e**) a scatter diagram of all the synthetic data and the whole original measured data, (**o**) the boxplots of the whole series. (**f**–**j**) The amplified diagram of the area enclosed by red lines in a scatter diagram.

**Figure 14 ijerph-16-04345-f014:**
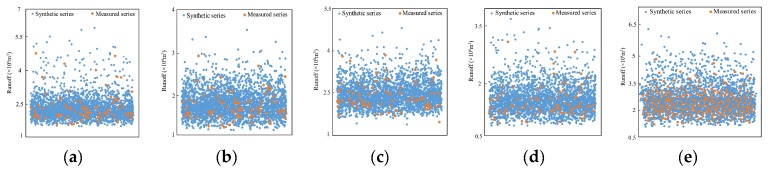
Scatter diagram (**a**–**d**) and boxplots (**f**–**i**) of the synthetic series and measured series in four segments at Gaojiabao gauge; (**e**) a scatter diagram of all the synthetic data and the whole original measured data; (**j**) the boxplots of the whole series.

**Figure 15 ijerph-16-04345-f015:**
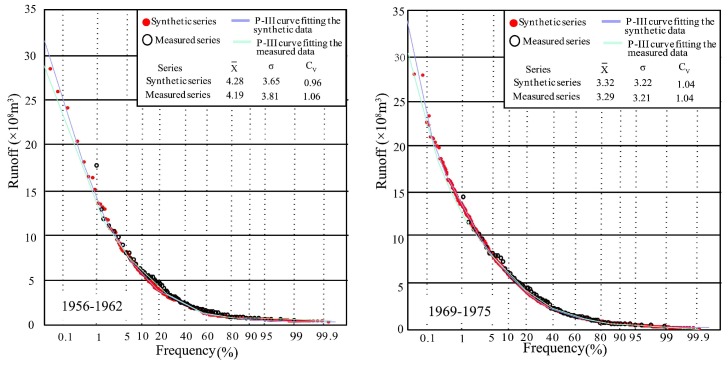
P-III frequency curves fitting segmented synthetic and measured runoff series and statistical parameters at Xianyang gauge.

**Figure 16 ijerph-16-04345-f016:**
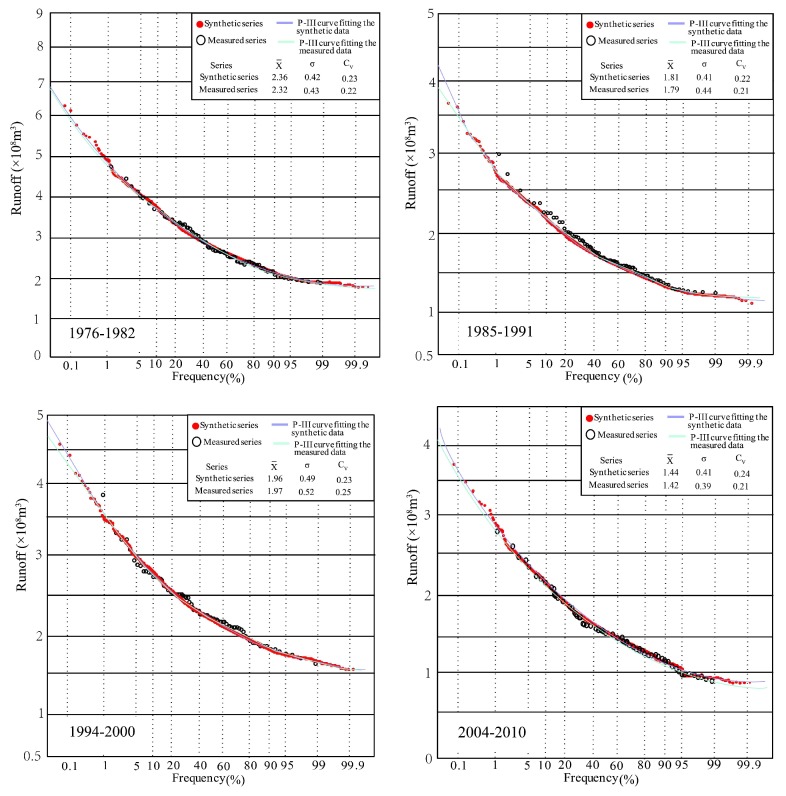
P-III frequency curves fitting segmented synthetic and measured runoff series and statistical parameters at Gaojiabao gauge.

**Figure 17 ijerph-16-04345-f017:**
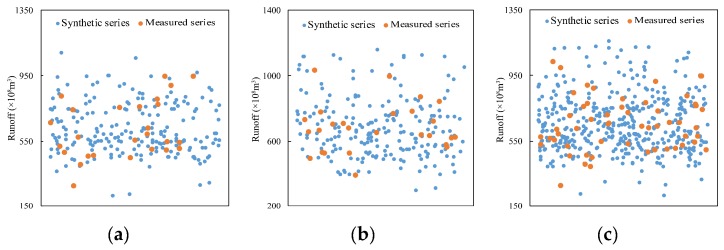
Scatter diagram (**a**,**b**) and boxplots (**d**,**e**) of synthetic series and measured series in two segments at Xiangtan gauge; (**c**) a scatter diagram of all the synthetic data and the whole original measured data; (**f**) the boxplots of the whole series.

**Figure 18 ijerph-16-04345-f018:**
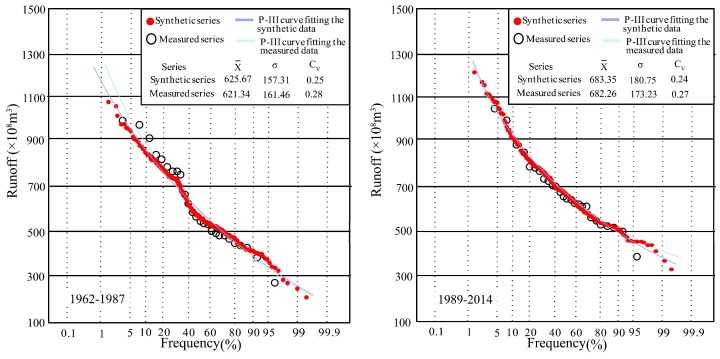
P-III frequency curves fitting segmented synthetic and measured runoff series and statistical parameters at Xiangtan gauge.

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
