# Peer review of "A Cross-Reconstruction Method for Step-Changed Runoff Series to Implement Frequency Analysis under Changing Environment"

_ijerph, 2019, doi:10.3390/ijerph16224345_

Round 1
Reviewer 1 Report
Main comments:
Authors should make the aim of the work more visible. “The aim of the study was to develop a new method ...” The description of the symbols in Formula 4 and 5 must be completed. I believe that the colour scale in Figure 5 was chosen incorrectly, usually the higher values are shown as red and the lower values as green. Authors should add a schema in which the stages of analysis should be presented. Chapter 4.2.1 should be better explained why such a segmentation was used (for Xianyang gauge 1956-1962; 1969-1975; 1987-1993; 1994-2000 instead of 1956-1967; 1969-1984; 1986-1992; 1994-2000). What should be done if it is not possible to create equal size series? The methodology should clarify in detail whether segmentation is performed on average annual flows and decomposition on average monthly flows. Line 287-288: how do you finally understand this phrase? "According to the locations of change points in mean and variance" in the case of Xianyang the Pettitt method shows three points and the ICSS method one point? In that case, choose the Pettitt method? What if the results were different from the Pettitt method and the ICSS method? Line 314-315: Authors indicate "And the residuals in various segments are relatively stable although there are small trend changes in segments". Please give some comment on whether the trend is right or wrong? Is the trend significant or not, what to do if the trend is statistically significant? I believe that in the section discussion should discuss its own results with those obtained by other researchers, now the section discussion is an auto-discussion.
Specific comments:
Line 48: Milly et al. change to Milly et al. [4] Line 53: Matalas change to Matalas [5] Line 56: Chung et al. change to Chung et al. [2] Line 84: Cannon et al. is should be checked [20,21] Line 85: Sarhadi et al. change to Sarhadi et al. [22] Line 86: Brodie change to Brodie [23] Line 87: Nasri change to Nasri [9] Line 88: Jiang et al. change to Jiang et al. [24] Line 91: change to Sklar [25] Explain what MSE, C.V. and C.S. in Figure 1 means. Line 135: Pettitt (1979) change to Pettitt [34] Line 153-154: Tiao (1994) [36] change to Tiao [36] Line 195: K-S test the full description should be given firs time In Figure 5, the description should be changed Hydrological gauge to Gauge station Elevation value to Elevation m a.s.l. Explain what the rectangles and the circles mean in Figure 2. In the case of Figures 8, 9, 14 and 15 instead of a); b); c); and d) it is better to provide descriptions as in Figures 10 and 11. Line 332: Change 1994-200 to 1994-2000 Each single application (a), (b), (c) and (d) should start with a new bullet point.
Reviewer 2 Report
This article is fairly well-written. The intent of this study is okay. Some of the results may add to the existing knowledge. However, the following comments may enhance the quality of the manuscript:
The method (data expansion) mentioned in the paper title does not match very well with the cross-reconstruction method used in this study. All acronym names, such as “IMF” should be defined when first appear and be used thereafter. The abstract is too long and contains lots of general comments and methodology used in the study. The introduction section is too lengthy and includes too much literature review. What is the real meaning of “dead and alive of stationarity”? Is there any real “natural” runoff exist anymore? Does real “natural” runoff need to be studied and analyzed for planning and development purposes? The exact objective of this study has not been clearly stated in the introduction section. The inset curves in Figure 1 is too tiny to be recognized. Please define all the variables listed in all the equations. Please unify the use of “K” and “k” in equation 3. What is “K-S” test? Please briefly described the purpose and how to use this test. Besides, results of this test have not been displayed in detail in the results section. Why use a whole sub-section to validate the C-R method? Any good supporting reference citation to support the validity of this method? Use on hydrologic data analyses? Please unify the use of “Fig.” and “Figure”. Use “billion m3” instead of “108 m3”. Color red should be used for higher elevation instead of color green in Figure 5. Caption for Figure 12 is unclear. Is it “Pearson” type III or “Person” type III? How do the sub-sectioned results compare with the non-sectioned results? Without these comparisons, it is hard to convince the C-R method really works. Applying the same method on two basins which are located very close to each other is not convincing that this method will work on basins in a far away distance or in another country. This paper includes too many (49) unnecessary references. This is not a “review” article. Too many spacing, punctuation, capitalization, and most importantly English language errors have been detected.Author Response
Please see the attachment.

Round 2
Reviewer 1 Report
The Authors have responded to all the comments contained in the review.
It recommends the paper for publication in its present form.
Author Response
Thank you very much for your help in modifying this manuscript.
Reviewer 2 Report
My previous comments on:
Applying the same method on two basins which are located very close to each other is not convincing that this method will work on basins in a far away distance or in another country.
have not been properly addressed.
Besides, I recommend not to use "natural" runoff because it is not easy to find. Why not use "non-human induced" runoff.
